# A Review on Digitalization of CSR during the COVID-19 Pandemic in Indonesia: Opportunities and Challenges

Enjang Pera Irawan *, Suwandi Sumartias, Soeganda Priyatna and Agus Rahmat

Faculty of Communication Science, Universitas Padjadjaran, Parahyangan 45363, Jawa Barat, Indonesia; suwandi.sumartias@unpad.ac.id (S.S.); soeganda.priyatna@gmail.com (S.P.); agus.rahmat@unpad.ac.id (A.R.)
* Correspondence: enjang19002@mail.unpad.ac.id

**Abstract:** The COVID-19 pandemic has become a global problem since first appearing in 2020. Not only does it heavily affect the health sector, but it also spreads to other sectors such as social, economic, and education. Studies have shown that many global companies, including those based in Indonesia, contribute to the global pandemic mitigation by implementing Corporate Social Responsibility (CSR) programs. So far, the implementation of CSR is mainly focused on providing food, medicines, and vitamins, as well as medical facilities and equipment. On the other hand, other reviewed studies showed that the pandemic has transformed the CSR implementation from offline to online, also known as CSR digitalization. The limitation in mobility and strict social distancing rules by the government have resulted in this emergence of CSR digitalization initiatives. Although CSR digitalization is still relatively rare, several technology companies have started implementing it. CSR digitalization practices aim to empower micro, small, and medium enterprises (MSME) to master digital competencies and increase their economic condition affected by the pandemic. Companies implementing CSR digitalization reported a more efficient and effective CSR implementation. This article can potentially introduce a new paradigm to the industry players on the importance of CSR digitalization and future opportunities due to the changes in the behavior of society post-pandemic.

**Keywords:** CSR digitalization; COVID-19; challenge; opportunity; Indonesia

## 1. Introduction

The COVID-19 pandemic has created a disturbance in our global society in a way that has never happened before. The vast impact of this pandemic has potentially changed our society's viewpoint, way of thinking, and livelihoods (He and Harris 2020). Most people and communities were impacted and are experiencing increased anxiety, stress, and panic (e.g., in a consumer society a phenomenon of 'panic buying' is identified (Anderson and Bedford 2020; Depoux et al. 2020; Saleh and Mujahiddin 2020)). This situation has affected people's physical and mental well-being and economic stability and created immeasurable risks for global health and the economy (Bretscher et al. 2020).

According to the Worldometer website, as of 21 January 2021 COVID-19 has infected 97.2 million people globally with 71.8% (69.8 million) recovered, and 2.1% (2.1 million) died (Arnani 2021). During the same time in Indonesia, the Indonesian COVID-19 Response Acceleration Task Force (Satgas COVID-19) has recorded over 951,651 cases with 81.2% recovered and 2.8% died (Aziz 2021). The survey conducted by the Indonesian Institute of Sciences (LIPI) between 24 April and 2 May 2020 reported that the pandemic led to the dismissal of over 15.6% of Indonesians from their jobs; 13.8% without severance pay. Most of these laid-off workers were young people (i.e., 15–24 years old). Several of the worst-affected sectors that need special attention were construction (29.3%), trading, restaurant, and services (28.9%), as well as transportation, warehousing, and communication (26.4%) (Meilianna and Purba 2020).

Governments made several interventions worldwide to mitigate the pandemic impact by suppressing the curve of COVID-19 cases (Shammi et al. 2020). Efforts were taken

to improve the currently overwhelmed healthcare system. Nevertheless, new cases kept emerging and the worldwide exposure has also increased (McKibbin and Fernando 2020). This situation has escalated into social conflict, since there was a limited supply of basic needs (e.g., food), increased commodity prices, and the loss of jobs (Tayo et al. 2021).

Various elements of our society have also participated in mitigating this pandemic, including within the private sector through companies' corporate social responsibility (CSR). Several companies in the US performed their CSR programs based on health, economic, and social needs emerging from the pandemic (Aguinis et al. 2020). In the UK, most large companies concentrated on pandemic mitigation through donations and philanthropic works (He and Harris 2020). In Arabic countries, the employees' security and safety during work were guaranteed by equipping them with protective COVID-19 gear (Al-nawafah et al. 2020). A lot of the world's best companies helped with coping with the pandemic crisis by reducing their employees' workload without reducing their rights, offering relief to their consumers, hosting business virtual-based activities, and donating equipment for COVID-19 relief and consumable products, as well as manufacturing medical equipment to be donated (Marom and Lussier 2020).

In Indonesia, the COVID-19 pandemic has impacted the health aspect of its society and hindered the national economy. However, the participation rate of Indonesian companies in CSR has also increased. For instance, there was a significant increase in participants in the TOP CSR Award 2020 event this year (120 finalists out of 200 companies) compared with last year (72 finalists out of 150 companies). The *TopBusiness* magazine initiated this event with the National Committee on Governance Policy (KNKG) and several CSR associations, businesses, and GCG, CSR consultant companies, and other nationally well-known businesses. The 2020 CSR Awards theme was the CSR strategic role in the national economic recovery in the new normal era (Ajang Bergengsi Top CSR 2020).

In Indonesia, state-owned enterprises (BUMN) and private companies alike, from online-based transportation, household product manufacturers, property developers, logistic deliveries, pharmaceuticals, and foreign companies are involved in mitigating the pandemic by providing logistics such as food, medicines and vitamins, and medical equipment and gear (e.g., masks, gloves, ventilators, protective gear, etc.,) (Lukman et al. 2019). Some companies, such as a few BUMN and private companies, help in alleviating the economic impact of COVID-19 through empowerment programs and digital business training designated for societies and small-scale business owners (Arifin et al. 2020).

Studies have shown that companies tend to get involved in mitigating the health impact more than the economic impact. However, CSR oriented in solving economic impact has a higher acceptance by society, since there are still limited companies working on this. Besides, the pandemic has changed consumer purchasing behavior from offline to online; thus, this condition can favor micro, small, and medium enterprises (MSME), who market their product digitally (Ibn-Mohammed et al. 2021). The CSR digitalization implementation will impact the MSME and companies, since society depends more on technology post-pandemic. Therefore, digitally oriented CSR will be more beneficial to society in the long run.

The COVID-19 pandemic is a momentum for companies to push the Indonesian digital economy through CSR digitalization. CSR digitalization is a method companies use to equip their CSR activity or programs with digital technology (Parente 2020). The pandemic has triggered a transformation wave from conventional businesses to online businesses. This is an exciting opportunity for Indonesia and other developing countries to increase their competitiveness. A report by McKinsey mentioned that the shift in the trading industry to the digital market will increase the economic growth up to USD 150 billion in 2025 (Das et al. 2016). The situation mentioned above challenges the improvement of the Indonesian digital-based economy, since the employment rate in the information and economic sector is relatively low (0.72%) and occupies the fourth lowest of the 17 main available employment areas (Badan Pusat Statistik 2020).

In this study, we discuss several early ideas on how the pandemic has affected the CSR implementation from offline to online; a process known as CSR digitalization. Literature study results revealed how the world's companies, especially in Indonesia, have participated in pandemic mitigation through CSR digitalization. In this pandemic era, we are more interested in exploring how the companies empower society and MSME through their digital-based CSR, and also in exploring opportunities and challenges within. This study seeks to provide new insight and alternatives to industry players on the importance of CSR digitalization during the pandemic and in the future.

## 2. Health, Economic, and Socio-Economic Impact of the COVID-19 Pandemic in the World and Indonesia

The unpredicted COVID-19 pandemic has shaken the world and currently there is no exact formula or planning on how to cope rapidly with the crisis (Pinner et al. 2020). Besides the health problems, this pandemic has disrupted the global economy, forcing the government, companies, and individuals to adjust properly (Sarkis et al. 2020; Sohrabi et al. 2020). This situation has revealed the weakness of the over-centralized global production and supply chain, which has resulted in a fragile global economy and a weak inter-industrial relationship (Bachman 2020; Sarkis et al. 2020). This situation has led to a high unemployment rate and increased food insecurity for millions of people because of the lock and restriction (Guerrieri et al. 2020).

Previous studies (Kulachinskaya et al. 2020) have explained several risks associated with this pandemic, e.g., health, economic, and social, as illustrated in Table 1. The health crisis has quickly triggered a financial crisis at the business level and a decreasing trend in the development of several industries (Bachman 2020).

**Table 1.** Health, economic, and social risks from the COVID-19.

| Health Risks | Economic Risks at the Business Level | Socioeconomic Risks |
|---|---|---|
| Absence of a vaccine or treatment against the disease | New security and cybersecurity risks for employees and customers | Unemployment, loss of income, and the emergence of situations of vulnerability |
| Insufficient stock of sanitary material and protective equipment | Operational risks due to limitation of face-to-face economic activity | Massive appearance of psychological problems due to pain from the loss of loved ones or from problems associated with the new personal and work situation, feelings of loneliness |
| Insufficient stock of hospital areas for seriously ill patients and the absence of facilities to house mild and asymptomatic patients, unaffected elderly, and other vulnerable groups | Operating costs not correlated with income | Need for training and leisure activities at home due to limited mobility |
| | Liquidity problems | |
| | Survival | |

These identified impacts also potentially transpire in Indonesia. The following section will explain various data and facts based on our literature study that explain the condition and situation resulting from the COVID-19 pandemic on health, economic, and social aspects in Indonesia.

### 2.1. Health Impact

In reality the COVID-19 pandemic has impacted individuals' physical and psychological conditions. It has caused deaths, reductions in physical fitness, and psychological effects, e.g., post-traumatic stress, confusion, anxiety, frustration, fear of affection, insomnia, and helplessness (Brooks et al. 2020). The worst condition is the emergence of xenophobia and suicidal cases due to an intense fear of the deadly virus infection (Fitria et al. 2020).

### 2.2. Economic Impact

The current pandemic situation is unique, where our mobility is limited, thus disturbing our activities. Everyone is focusing on minimizing their interaction to avoid spreading this virus. To prevent COVID-19 virus from spreading, most countries closed their borders, limited their citizens' movements, and even quarantined their citizens inside their houses for weeks. Many people felt intense pressure and lost their jobs, because the slow economic situation caused them to go bankrupt. This economic condition has never been seen since the Great Depression in the 1930s (Donthu and Gustafsson 2020).

This current situation has affected the macro-economy in countries worldwide. For example, it halted the production of various basic-need products and the export and import activity of these products (Iyengar et al. 2020). On the micro-scale, the economic impact listed was the changes in the consumers' behavior such as panic buying (Sim et al. 2020) and focusing on buying essential basic needs since other products were scarce (Spash 2020).

In Indonesia, up to 2.6 million people have lost their jobs due to COVID-19, leading to a rise in the national unemployment rate of 9.7 million people (Fajri 2021). In 2020, the economic growth fell −5.32% in the second trimester and −3.49% in the third trimester, leading to a recession trap in 2021 (Rachbini 2021). There are over 30 million MSMEs that cannot adapt yet to use digital technology during the pandemic (Darmawan 2021). Other sectors (e.g., tourism, transportation, export, and investor interest) have also been influenced (Nasution and Muda 2020), with the stock market deviating to a negative trajectory (Devi et al. 2020).

### 2.3. Socio-Economic Impact

Several socio-economic activities stopped due to the pandemic: millions of people were quarantined; borders closed; schools closed; automotive industry, airlines, as well as transportation companies collapsed; trading exhibition, sports events, entertainments events canceled; millions became unemployed; international tourism locations abandoned; nationalism and protectionism appeared (Basilaia and Kvavadze 2020).

A survey-based study in 2021 (SMERU et al. 2021) identified at least five impacts of the COVID-19 pandemic experienced by Indonesian society:

1. Decrease in household income. Survey results showed that three out of four people had experienced a reduction in their income. One out of two changed their employment from formal workers to informal occupations. One out of two also did not have savings to support them, and nine out of ten family businesses were impacted.
2. Children experienced learning loss. There were 75.3% of households with children with reduced income, resulting in 2.1 million children living in poverty. Children experienced problems in learning such as limited access to a good internet connection for their remote learning (57%), and almost three out of four parents were concerned about this matter. In the health aspect, 40% of parents feared that their children would be infected and about 13% of children under the age of five were not vaccinated. Other risks were reported, e.g., almost 45% of parents experienced behavioral challenges from their children, 7% of parents had at least one child working, and 2.5% of that number already worked since the pandemic.
3. Women experienced an increase in responsibilities and chores in parenting. They had difficulties keeping the balance between household chores and other additional responsibilities of children's remote learning.
4. Food security and vulnerable groups have to be prioritized in the future. Almost 30% of respondents worried they could not feed their families. The proportion of households with medium to severe food insecurities rose to 11.7% in 2020. Decreased income and the disturbance in the food supply system were the primary factors causing food insecurities.

In essence, the study showed that the COVID-19 pandemic affected health, economy, and social aspects. However, as a part of the world community we should cooperate and help minimize the virus spread by obeying health protocols established by the government.

In the broader scope, society hopes that all elements, especially big companies, could help the government and mitigate the pandemic effects through its own CSR. Hopefully, through the involvement of all elements the pandemic situation will be resolved and get back to normal once again.

The COVID-19 pandemic is a momentum for the industry players to help the government in recovering the national economy, considering that the latter owns more resources in hand; however, the participation level of industry players to recover the national economy through CSR is relatively low. CSR digitalization enables the industry players to support economic recovery and national health. Industry players can give online training to empower society, especially MSMEs, and still comply with national health protocols. Society can create and market its products online with digital competencies while minimizing risky online activities.

## 3. An Overview of CSR in Indonesia: Before and during the COVID-19 Pandemic

### 3.1. CSR before the COVID-19 Pandemic

Since the pandemic began, the CSR implementation trends have changed. Before the pandemic, CSR programs relied on the companies' missions, cultures, environments, risk profiles, and operational conditions (Akbar and Humaedi 2020). In this review, the overview of different CSR refers to six main CSR activities as explained by Kotler and Lee (2005), e.g., cause promotions, cause related marketing, corporate social marketing, corporate philanthropy, community volunteering, and socially responsible business practices (Kotler et al. 2012). Through this, we will describe a trend overview of CSR before and after the pandemic.

An example of a cause promotion CSR program that aims to encourage people to donate is executed by the Danone company. Danone established a program called "one for ten". Every purchase of one liter of their water bottle product means ten liters of water will be donated to the people in East Nusa Tenggara, Indonesia (Inovasi Danone-AQUA 2019). A corporate social marketing CSR program shown by PT Unilever Indonesia, which aims to change society's behavior for the better, launched the healthy hands program or the national campaign to wash hands with soap (Nurbaiti et al. 2020). Surabaya Patata showed a cause related marketing CSR program. They aim to donate a part of their profit to the ex-prostitution area, colloquially known as "Dolly", in Surabaya, Indonesia (Ferdiyanti and Dyatmika 2019).

Several companies launched the corporate philanthropy CSR program. For example, one of the big telecommunication companies in Indonesia, XL Axiata, donates internet data to schools (Yasa 2017), PT Antam Tbk donates food for the flood disaster survivals in North Kolaka (GATENEWS.ID 2020), while PT Djarum gives 6000 scholarships through their program Djarum Scholarship Plus. The community volunteering CSR programs are also implemented by several companies such as PT Astra International Tbk through their Astra Employee Volunteer program, which involves their employees cleaning bus stations in Jakarta (Setiawan 2018), and PT PLN (Persero), through their Employee Volunteer Program (EVP) program, planted trees, gave training on recycling garbage, and educated students on electricity (O&G Indonesia 2015). PT Freeport Indonesia executed the socially responsible business practice CSR through their 3-R (Reuse, Reduce, and Recycle) program in all of their operational areas (Freeport Indonesia 2016) and Bukalapak company with their Paperless Program (Sujanto 2018).

Before the pandemic, several CSR programs directly helped people by building the infrastructure of social and religious facilities (Huda 2019), donating to education facilities, giving scholarships (Apriatma et al. 2019), donating to survivors of natural disasters (Ramadhan and Wijaya 2020), and others.

### 3.2. CSR during the COVID-19 Pandemic

The CSR trend activities changed along with the widespread impact of the COVID-19 pandemic. Studies showed that companies worldwide prioritized their CSR for pandemic

prevention and handling (Aguinis et al. 2020; He and Harris 2020; Marom and Lussier 2020). Besides that, the pandemic mitigation in countries with severe cases was carried out through non-pharmaceutical interventions such as travel restrictions, border closures, event cancellations, lockdowns, social distancing, school/workplace closures, and social gathering prohibitions (Chiodini 2020). Studies showed that this pandemic could be viewed as an opportunity for public and private sectors to show their commitment and participation in protecting citizens through implementing their CSR in solving problems caused by the pandemic (Al-nawafah et al. 2020).

The conditions mentioned also apply in Indonesia, where companies added CSR programs that aimed at mitigating pandemic-related health problems. Those programs helped to relieve the health, economic, and socio-economic problems. A summary of several CSR activities representing companies from different sectors, e.g., transportation, oil and gas, household production, property, electrical, pharmacy, logistic services, banking, and telecommunication is presented in the following table (Table 2). Do note that the data are only a representation of the actual data. In reality there are presumably more companies that practice their CSR activities in Indonesia.

**Table 2.** CSR activities from selected companies in Indonesia during the COVID-19 pandemic.

| Health Mitigation | | Economic Mitigation | | Socio-Economic Mitigation | |
|---|---|---|---|---|---|
| **Program** | **Company** | **Program** | **Company** | **Program** | **Company** |
| Aid such as food, medicine and vitamins, medical tools and equipment (masks, sanitizers, gloves, ventilators, personal protective equipment, etc.) | - GRAB<br>- PT Blue Bird Tbk<br>- PT. Pertamina<br>- PT Chevron Pacific Indonesia (CPI)<br>- WINGSGROUP<br>- SINARMAS LAND<br>- SINARMAS LAND<br>- PT. Perusahaan Listrik Negara (PLN)<br>- PT Kalbe Farma<br>- PT Industri Jamu Dan Farmasi Sido Muncul Tbk<br>- JNE<br>- BCA<br>- PT Telkom Indonesia<br>- Indosat Ooredoo<br>- PT XL Axiata Tbk | Training on digital business for communities and SMEs | - PT Perusahaan Listrik Negara (PLN)<br>- PT. Pertamina<br>- Indosat Ooredoo<br>- PT XL Axiata Tbk<br>- APP Sinar Mas<br>- Tokopedia | Digital literation for the society | - GRAB<br>- PT Blue Bird Tbk<br>- PT. Pertamina<br>- PT Chevron Pacific Indonesia (CPI)<br>- WINGSGROUP<br>- SINARMAS LAND<br>- SINARMAS LAND<br>- PT. Perusahaan Listrik Negara (PLN)<br>- PT Kalbe Farma<br>- PT Industri Jamu Dan Farmasi Sido Muncul Tbk<br>- JNE<br>- BCA<br>- PT Telkom Indonesia<br>- Indosat Ooredoo<br>- PT XL Axiata Tbk |

This condition also applies in Indonesia, where companies added CSR programs that are aimed at helping pandemic mitigation in the health sector through the donation of

food, medicine, vitamins, health equipment (e.g., masks, ventilators, gloves, sanitizer), and protective gear. These companies came from various sectors: transportation such as GRAB (Triana et al. 2020) and PT Blue Bird Tbk (Gunawan 2020); oil and gas such as PT Pertamina (Muna et al. 2020) and PT Chevron Pacific Indonesia (CPI) (Lusiana 2020); household manufacturer WINGSGROUP (Triana et al. 2020); property developer SINARMAS LAND (Triana et al. 2020); electricity provider PT Perusahaan Listrik Negara (PLN) (Anggoro 2020); pharmaceutical such as PT Kalbe Farma (KALBE 2020) and PT Industri Jamu dan Farmasi Sido Muncul Tbk (Lukman et al. 2019); logistic courier JNE (Mayasari 2020); banking BCA (Ulya 2020); and telecommunication such as PT Telkom Indonesia (Santoso 2020), Indosat Ooredoo (Damar 2020), and PT XL Axiata Tbk (Wardani 2020).

Several companies have implemented CSR programs that are meant to help mitigate the economic impact of the pandemic. Such programs were digital training targeted on the society and SMEs by PT Perusahaan Listrik Negara (PLN) (Arifin et al. 2020), PT PERTAMINA (Pertamina 2020), Indosat Ooredo (Indotelko.com 2021), PT XL Axiata Tbk (Wardani 2020), PT Telkom Indonesia (Maarif 2020; Santoso 2020), APP Sinar Mas (Tribunnews.com 2021), and Tokopedia (Iskandar 2020). The program description will be explained in the following section of the CSR digitalization practices.

Industrial players realized the importance of CSR for their companies and the wider society. Besides helping the health aspect of the pandemic mitigation, CSR is also directed to help solve the economic problem by providing training on digital competence for society and increasing the digital marketing capability of MSMEs. Such competence is vital for the general public, especially those affected by the pandemic.

## 4. CSR Digitalization Practice during the COVID-19 Pandemic in Indonesia

The pandemic forced us to practice social distancing to prevent the spread of the virus. As mentioned before, this pandemic has weakened the economy, since it hampered the production and distribution processes. On the other hand, the mobility restriction has accelerated the digital platform adoption process for many things (Puriwat and Tripopsakul 2020); thus, people will develop a habit in the future to maximize the use of digital media for their needs (He and Harris 2020). This situation should trigger Indonesians to adapt fast to digital technology development to increase their productivity in the future.

The contributions of companies through their CSR programs were also expanded into the Indonesian economic recovery. Those programs comprised digital business training targeted at the public, digital marketing training for MSME, and other training aimed to increase the public capacity on exploiting digital technology as a source of income during the pandemic.

There are several companies committed to this type of CSR. For instance, state-owned enterprises such as PT Perusahaan Listrik Negara (PLN) and PT PERTAMINA held digital marketing trainings for MSME (Arifin et al. 2020). Private companies such as Indosat Ooredo and PT XL Axiata Tbk also held digital competence training for the millennial generation and MSME (Wardani 2020), while PT Telkom Indonesia held training on digital literation for the public (Santoso 2020). APP Sinar Mas held digital marketplace training for MSME (Tribunnews.com 2021) and Tokopedia held MSME go digital training (Iskandar 2020).

Digital-based CSR practices by Indonesian companies mainly focused on giving digital economic competence, either for society or MSME. Such events are why digital literation is crucial for choosing, understanding, and using according to different needs (Santos and Serpa 2017). To benefit from digital technology, literation and competencies need to be improved. Digital competence can be defined as using ICT and digital media to execute tasks, solve problems, convey, organize information, collaborate, produce and share content, and develop knowledge effectively (Klassen 2019).

Digital competence training educates the public to produce digital content to support their economy (López-Meneses et al. 2020). This implies the vital skill of critically using digital technology (Klassen 2019). Digital competence covers the skill to use digital technology, either as a consumer or a content maker, in public or private life (Feng 2020). Therefore,

digital literation and competence are essential to be promoted and widely introduced to society (Ramsetty and Adams 2020).

From the studies mentioned, it can be concluded that CSR digitalization practices have enabled the recipient community to become more empowered and independent. In the current pandemic situation, communities, especially MSMEs, desperately need digital competences to survive. Physical activities can be converted online by gaining digital competencies, such as executing marketing and transactions. In addition, people who work in offices can still work using digital technology. Therefore, CSR digitalization can minimize additional risks from the pandemic impact. Furthermore, CSR digitalization based on society empowerment has been considered more effective and efficient, also reducing the society's dependency on help from companies.

## 5. Opportunities and Challenges on CSR Digitalization during the COVID-19 Pandemic

### 5.1. Opportunities in CSR Digitalization

The COVID-19 pandemic has posed significant consequences on social life, economy, and culture all around the globe, especially on society's habit of using digital technology (Melentieva 2020). Before the pandemic, digital technology was one of the pragmatic steps in helping human activities to be more efficient and effective (Gladden 2019). During the pandemic, realized or not, the use of digital and online platforms has increased gradually and become a part of our lifestyle (Puriwat and Tripopsakul 2020).

The pandemic has decreased the Indonesian consumer's capacity, since many people lost their jobs and most have reduced their interaction by staying at home (Djumena 2020). Many MSMEs that were not selling their products online have closed their shops due to this situation (Purnomo 2019). Nevertheless, this situation has revived the MSMEs to change their selling strategy to online; thus, marketing competence in social media and the marketplace is needed (Arianto 2020). This online business transformation in Indonesia can increase its competitiveness at the national and international levels (Pandemi COVID-19 2020).

During this pandemic, many occupations were forced to close down, such as mid-level managers in the financing sector, consultants, and administrators in the private sector. On the other hand, the digital-based occupation has become the top ten most demanded position in the job market (Dewi et al. 2020). Thus, digital technology in marketing strategies such as CSR needs to be highly considered (He and Harris 2020). People are hopeful that corporations can increase their digital competencies by realizing CSR.

This condition has triggered increased CSR attention and demand worldwide, especially in Indonesia (Nugroho 2020). Implementing CSR is a momentum for companies to support good relations with their stakeholders and gain legitimation and reputation (Lanis and Richardson 2012). For the company to survive and grow, it needs to pay attention to the needs of various stakeholders and legitimize activity to maintain the conformity between the purpose of society and the company (Frynas and Yamahaki 2016).

Despite the motive difference, implementing digital-based CSR is currently critical. Studies have shown that digital technology has been effectively helping human activities (Daniel 2020), potentially becoming an important element in reducing the digital disparity (Mavrou et al. 2017), and this current condition can change the global economy order towards digital economy and finance (Kickbusch et al. 2005). Through digital competence, individual participation in the development will increase, regardless of his/her varied background (e.g., social status, age, sex, and disability) (Fraile et al. 2018). It is predicted that even after the pandemic is over, the use of digital technology will keep increasing to overcome the disturbance in the supply of goods due to social distancing (Pagoropoulos et al. 2017).

At this moment, the public hopes that companies could use digital media to communicate CSR through dialog in social media or other digital media (Kent and Taylor 2016). They can use this online CSR dialog as a means of communication and decision-making (Illia et al. 2017).

Based on previous studies, companies that successfully execute CSR digitalization potentially gain benefits, such as reaching a broader target (Janani and Gayathri 2019), gaining new opportunities to participate in an open dialog with the public, and simply having better communication with the public (Illia et al. 2017). A study by Borger and Kruglianskas (2006) summarizes that digital technology has a positive effect on a company's progress (Borger and Kruglianskas 2006).

Recently, a technology-based CSR program was launched by Google and positively appreciated by investors, users, stakeholders, and others (Du et al. 2010). A study by Lee et al. (2013) on 500 companies, who communicate their CSR program online and on social media, mentioned that only 222 companies with excellent performance are willing to communicate their CSR program and gain public appreciation (Lee et al. 2013). Companies with excellent performance and CSR-awarded were the ones that communicated their achievements to the public on social media and other online-based media (Ahmed 2016). Young consumers pay more attention to the company's social media or other digital media to judge the company's performance (Wella and Chairy 2020).

Stakeholders also considered CSR digitalization as important. They expect their companies to adopt digital technology to communicate CSR to the public and themselves (Colleoni 2013) in a simple and easily understood manner (Kesavan et al. 2013). A company that sincerely executes its CSR appears more reliable and has an excellent reputation from the stakeholder's perspective. Besides that, CSR implementation has a vital role in the company's risk management (Eisenegger and Schranz 2011). The summary of the opportunities for executing digital-based CSR in line with society's needs can be seen in Figure 1.

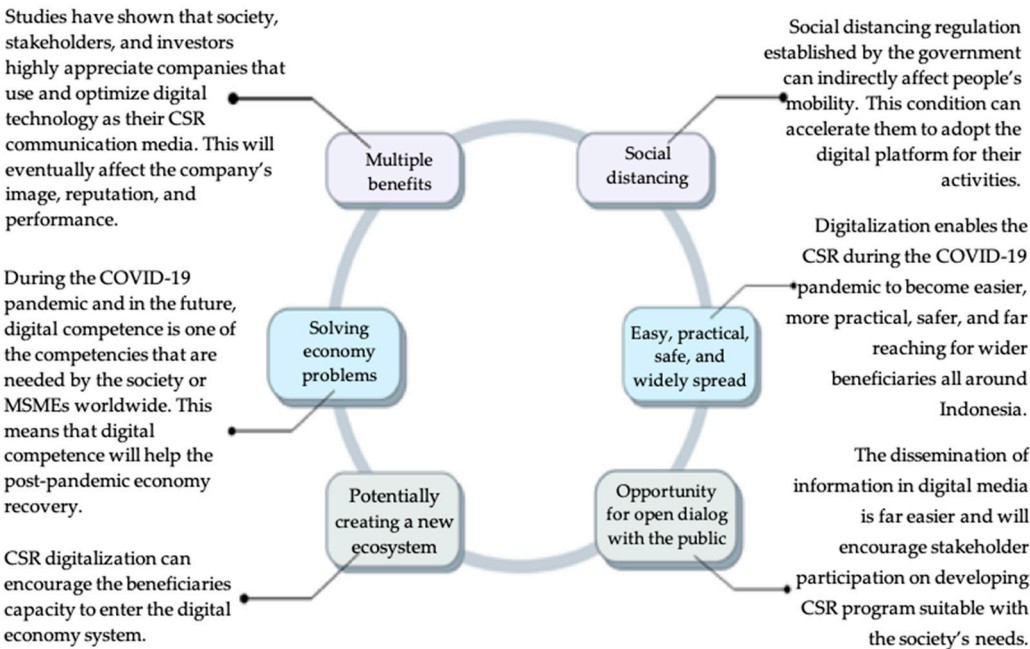

**Figure 1.** Opportunities of CSR digitalization implementation.

Companies that are carefully looking into the opportunities of CSR digitalization during the COVID-19 pandemic could potentially acquire benefits for themselves and their beneficiaries. The CSR digitalization step is very crucial, since it is highly relevant to the current situation and future trends. Therefore, hopefully the analysis and findings of this study can motivate companies to present digital-based CSR to the public during this time of crisis.

Undeniably, the existence of a company is not enough just by paying tax and obeying the law. They also need to contribute to society through their CSR program (Kourula and Delalieux 2016). Philosophically, CSR can be seen as a responsible act that allows companies

to achieve their competitive excellence sustainable by all means (McWilliams and Siegel 2011). The World Business Council for Sustainable Development (WBCSD) defines CSR as the business player's commitment to behave ethically and to contribute to the economic development and life quality of their workers and family, the local community, and broader society. A more ethical understanding of CSR is an act of the company to manifest social kindness, more than just efforts to obey law and order or to relieve the pressure from their stakeholders (Torea et al. 2020).

Generally, there are at least six reasons behind a company's motives to adopt CSR practices, e.g., innovation, expense reduction, brand differentiation, long-term consideration, and the involvement of customers and employees. Other benefits are: increased sales and income, market expansion, a better working environment, better relations with the local authorities, and increased crisis management. Studies have shown that CSR behavior has positively affected individual performance, organization attitude, and organization attractiveness (Farooq et al. 2017); increasing the company's reputation in the perspective of society and the nation, as well as better employees' involvement and relations (Rupp et al. 2013).

Studies show that companies and their stakeholders' civil involvement and cooperation can build social capital. The company's potential resistance and risk can also be consequently anticipated (Dunbar et al. 2020). The social capital built through CSR activities could reduce the asymmetrical information with stakeholders (Li et al. 2019). The convenience gained through CSR digitalization enables a more straightforward, efficient, and effective collaboration. Previous studies (Devin and Lane 2014; Girard and Sobczak 2012; O'Riordan and Fairbrass 2014) showed that many companies that adopted an idea to involve the public in their CSR initiatives were more successful.

Cohen and Prusak (2001) mentioned that social capital is a factor that can facilitate the development of society. The role of social capital is essential to increase the quality of cooperation in society. Social capital can grow the network, trust, and social relation elements to support CSR continuity. Adopting social capital by implementing CSR means that the company and society bear program continuity.

*5.2. Challenges of CSR Digitalization*

This review has identified several potential challenges to CSR digitalization. For example, the operational level of the company will have to work from home (WFH) through the social distancing policy set by the government. All CSR programs will have to be done virtually by the CSR team at their own home. This situation is challenging since the company must adapt quickly to coordinate and to communicate virtually with the internal team, stakeholders, and beneficiaries. Indeed, this can give a unique challenge for all parties involved.

Although challenging, several companies successfully adapted to the pandemic with limited mobility due to social distancing (Masters et al. 2020). The geographical condition of the Indonesian archipelago with some relatively difficult areas to access is also another challenge. Here, the CSR program can be communicated through virtual media optimization such as the Zoom meeting application, the *WhatsApp* application, social media, and the company's website (Website Training Online 2020). However, there are still a lot of areas in Indonesia without internet access coverage since telecommunication infrastructure is constrained in remote areas.

A study by Janani and Gayathri (2019) further mentioned six challenges of implementing digital-based CSR, e.g., data available online from the stakeholders cannot be trusted completely, it is difficult for remote areas to accept CSR, additional budget is needed to educate people in remote areas to use technology, the digital-based CSR cannot solve several village problems, the possibility of the company to violate the user's privacy and data in the name of CSR, and difficulties in controlling fake information on CSR that can easily spread (Janani and Gayathri 2019). The summary of possible potential challenges of implementing digital-based CSR can be seen in Figure 2.

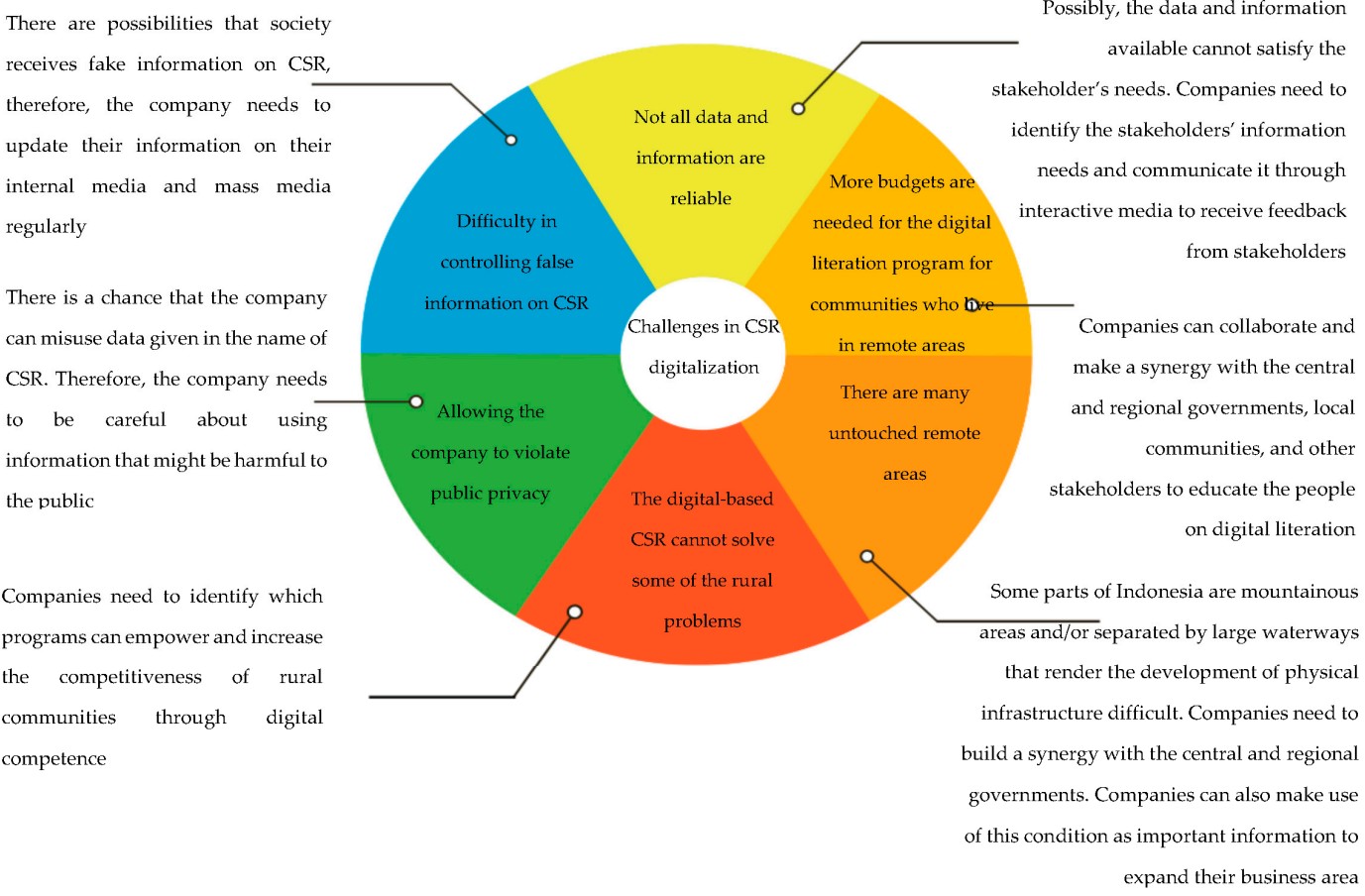

There are possibilities that society receives fake information on CSR, therefore, the company needs to update their information on their internal media and mass media regularly

There is a chance that the company can misuse data given in the name of CSR. Therefore, the company needs to be careful about using information that might be harmful to the public

Companies need to identify which programs can empower and increase the competitiveness of rural communities through digital competence

Possibly, the data and information available cannot satisfy the stakeholder's needs. Companies need to identify the stakeholders' information needs and communicate it through interactive media to receive feedback from stakeholders

Companies can collaborate and make a synergy with the central and regional governments, local communities, and other stakeholders to educate the people on digital literation

Some parts of Indonesia are mountainous areas and/or separated by large waterways that render the development of physical infrastructure difficult. Companies need to build a synergy with the central and regional governments. Companies can also make use of this condition as important information to expand their business area

**Figure 2.** Challenges of CSR digitalization implementation.

Facts from the East Ventures Digital Competitiveness Index (EV-DCI) 2020 report showed that the digital competition index in Indonesia is 27.9 on a scale of 0–100. This number implied that the Indonesian digital competitiveness was low. The sector with the lowest score was human resources and entrepreneurs, meaning that there are still limited human resources with skills in the digital economy. To increase the index, entrepreneurs in the digital economy need to be improved. Thus, CSR digitalization is a movement that must be done consistently and sustainably. Through this it is possible that the Indonesian digital competitiveness index increases and supports the digital economic growth in Indonesia.

CSR digitalization collaboration with multiple stakeholders could result in an ecosystem that supports the digital economy. To achieve such conditions, numerous stakeholders' synergy, collaboration, and readiness are needed (Fadilah 2019, p. 18). For instance, the government/Indonesian Central Bank/regulator must support the development of a conducive ecosystem through developing instruments such as programs and policy along with central and regional regulations. The industry/business stakeholders must act as the economic driving force. Besides doing business activities programs are needed to help develop society's digital economy. Communities/societies need to actively utilize digital technology such as e-commerce. The university practitioner/professional must grow digitally taller.

Based on the aforementioned narrative CSR programs implemented by companies need to be synergized with government programs to make them target specific and suitable to the needs of society. Moreover, the government of developing countries, including Indonesia, actively supports economic growth through the digital sector. As predicted by McKinsey in their report, the collaboration and synergy between institutions can exponentially accelerate economic growth. The COVID-19 pandemic has opened a path for the acceleration of the digital economy. Through digitalization communities, especially

MSME with small capital, can still compete amidst the pandemic situation and in the future digital era.

## 6. Conclusions

COVID-19 has accelerated society's capacity to adopt digital platforms and catalyzed digital transformation. We suggest that companies or industry players widely adopt the digital platform to increase their business performance and to fulfill the stakeholders' hopes, including CSR. As proven by previous studies, the shift in the trading industry to the digital area will increase economic growth. Besides, companies that use digital media to implement their CSR tend to gain more public appreciation.

This study offered several early thoughts on how the COVID-19 pandemic is viewed as an opportunity instead of a problem. The pandemic is a momentum for companies to transform their conventional CSR to a digital platform; a process commonly known as CSR digitalization. CSR digitalization can be directed to programs based on the empowerment of society. During the pandemic, society, especially MSME, needs digital competencies to thrive. CSR digitalization programs are relevant at present and in the future. Companies have rated CSR based on their capacity to empower communities, noted as more effective and efficient and able to reduce society's dependencies on companies' philanthropic support.

Starting CSR digitalization is not easy and has been a challenging process. Companies need to consider the reliability of available data, the safety of data management, uneven access to technology in remote areas, budget allocation, and problems that CSR digitalization cannot solve. We understand the limitation of exploring those mentioned; thus, studies on alternative solutions from challenges in CSR digitalization need to be conducted in the future. Such studies can provide a guide for companies to start CSR digitalization. Industry players are suggested to keep innovating on efforts to develop the digitally oriented CSR by collaborating with the government, educational institutions, and communities focused on developing the society's digital competencies.

We conclude this study by asking CSR practitioners to collaborate with academics on planning and executing research-based CSR. Academics can identify possible opportunities and challenges and give alternative solutions relevant to the need of CSR implementors or beneficiaries. We also emphasize that the COVID-19 pandemic has strongly impacted all sectors of life; however, it also challenges us to be more innovative and solution-oriented.

**Author Contributions:** Conceptualization, E.P.I., S.S., S.P, A.R.; methodology, E.P.I., and S.S.; software, E.P.I.; formal analysis, E.P.I.; investigation, E.P.I., S.S.; resources, E.P.I., A.R.; data curation, E.P.I.; writing—original draft preparation, E.P.I. and S.S.; writing—review and editing, E.P.I.; supervision, S.S.; project administration, E.P.I..; funding acquisition, E.P.I. All authors have read and agreed to the published version of the manuscript.

**Funding:** This research received no external funding.

**Data Availability Statement:** The data presented in this study are available in the cited articles.

**Conflicts of Interest:** The authors declare no conflict of interest.

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
