# Peer review of "A Review on Digitalization of CSR during the COVID-19 Pandemic in Indonesia: Opportunities and Challenges"

_socsci, doi:10.3390/socsci11020072_

Round 1
Reviewer 1 Report
The article deals with topical issues. Everyone is writing about COVID 19 because it is an important topic today. However, in my opinion, the article is too general - it is a descriptive, review article.
So I believe - the article is written correctly, the structure is OK, but it is not a scientific consideration. What is missing is the scientific element - the "seed of science".
The connection with CSR in this form doesn't convince me. The conclusions drawn on the basis of the analyzes do not raise any new issues.
The offer of cooperation for other people / units is very good - and I think that this is the direction and after doing the activities you can write a scientific article that will be very interesting and scientific.
Author Response
Dear
reviewer
We are very happy to get input from you. For that we try to make improvements according to your suggestions.
The improvements we have made are as follows:
1. We have tried to add in-depth analysis, so it is hoped that we can master the scientific elements in this paper.
2. In conclusion, we have presented a new issue or problem that can be investigated by further researchers.
We realize that there are still weaknesses in this article, but we have tried to make improvements according to your feedback.
We wish you and your family good health and success.
Regards
Enjang Pera Irawan

Reviewer 2 Report
attached

Author Response
Dear Reviewers
We thank you for your valuable input to our journal. For that we have made some improvements according to your suggestions
- The contributions of the paper to the body of knowledge and current practice have been clarified and added extensively, particularly in the last paragraph of the Introduction section, last two paragraphs of the Discussion section, and the whole description of the Conclusion section
- Both Dunbar (2020) and Li et al. (2019) have been added in the discussion section and further elaborated to widen the scope of the discussion.
- The specific addition has been made, particularly in the last paragraph of the Introduction section, as well as addressed succinctly in the new version of the Abstract.
- The manuscript has been re-proofread, with specific attention to typographical and grammatical errors. Sentences in general have been made simpler and vague words have been replaced.
- All of the points recommended have succinctly been added to the current Conclusion. In general, we highlight that the study revealed how companies, through CSR, see Covid as an opportunity (rather than a problem) to accelerate the transition process towards digital economy.
This is the response we can convey. Wish you and your family always healthy and given success
Best Regard
Enjang Pera Irawan

Reviewer 3 Report
The paper provides the viewpoint on digitalization of CSR during COVID 19 in the country of Indonesia. The study has a country-specific element and the author should provide additional details and information about why this study can be beneficial to other developing countries or emerginng economies with similar problems or issues. The author offers the details of health, economic, and socioeconomic risks and impacts. However, the types of CSR dimensions are not clearly discussed in relation to risks and impacts. For this point, it should be more systematic and appropriate to present the the types of CSR dimensions and projects in digitalization in health, economic and socioeconomic aspects. Furthermore, the additional details of CSR projects should be also included, such as the owners of CSR projects and lengths of CSR projects. The part may be presented in tables in order to see the overall situtation in Indonesia.
The results of the study provided a useful discovery on CSR digitalization with figures and details. However, there is no appropriate discussion with the past or related research. The author should include more details concerning practical contributions or managerial implications from the study and directions for future research. The author offers the conclusion, but the contents of the conclusion was too limited and should be expanded for clearer understanding.
Author Response
Dear
reviewer
We are very happy to get input from you. For that we try to make improvements according to your suggestions.
The improvements we have made are as follows:
1. We have provided an analysis of the benefits of this article for developing countries, especially Indonesia.
2. We have added related examples of CSR digitization carried out by companies in Indonesia in health, economic, and socio-economic aspects. However, the limited time in making revisions made it difficult for us to collect detailed data, for that we only provide an overview. Hope you give tolerance.
3. We have explained related
practical contribution of this research. Then we also offer a more comprehensive conclusion. In addition, we have also provided suggestions and directions for further research in the future.
Regards
Enjang Pera Irawan

Round 2
Reviewer 1 Report
I appreciate the changes that have been made. I believe that everyone has their own perception of the topic.
In my opinion, the article is correct. The changes introduced show the commitment and determination of the authors.
Good luck.
Reviewer 2 Report
Well done. Congrats!